# Application of Maillard Reaction Products Derived Only from Enzymatically Hydrolyzed Sesame Meal to Enhance the Flavor and Oxidative Stability of Sesame Oil

**DOI:** 10.3390/molecules27248857

**Published:** 2022-12-13

**Authors:** Gang Ma, Shudong He, Shuyun Liu, Zuoyong Zhang, Tao Zhang, Lei Wang, Youshui Ma, Hanju Sun

**Affiliations:** 1School of Food and Biological Engineering, Engineering Research Center of Bio-Process of Ministry of Education, Key Laboratory for Agricultural Products Procssing of Anhui Province, Hefei University of Technology, Hefei 230009, China; 2Anhui Fengyang Yushan Grease Co., Ltd., Fengyang 233100, China

**Keywords:** sesame oil, sesame meal, Maillard reaction products, volatile compounds, oxidative stability

## Abstract

The low-temperature roasting of sesame oil has become increasingly popular because of its nutritional benefits; however, the flavor is reduced. In order to improve the quality of sesame oil without exogenous addition, sesame meal was hydrolyzed and further used to prepare Maillard reaction products (MRPs) while protease hydrolysis (PH) and glucoamylase-protease hydrolysis (GPH) were used, and their respective Maillard products (PHM and GPHM) were added in the oils for reducing sugar and total sugar content determination, free amino acid determination, and color and descriptive sensory analysis, as well as electronic nose, SPME-GC-MS, odor activity value, and oxidative stability analyses. Results showed that the MRPs could be produced using the enzymatically hydrolyzed sesame meal without exogenous addition, and the oil flavor blended with GPHM (GPHM-SO) was significantly (*p* < 0.05) improved with the best sensory quality. The composition of pyrazines (119.35 μg/mL), furans (13.95 μg/mL), and sulfur substances (6.25 μg/mL) contributed positively to sensory properties in GPHM-SO, and 2,5-dimethylpyrazine, 2,6-dimethylpyrazine, and 2,3-dimethylpyrazine were characterized as the key flavor compounds with odor activity values of 7.01, 14.80, and 31.38, respectively. Furthermore, the oxidative stability of the oil was significantly improved with the addition of MRPs, and the shelf life of GPHM-SO was predicted to be extended by 1.9 times more than that of the crude oil based on the accelerated oxidation fitting analysis. In general, the MRPs derived only from sesame meal can enhance the flavor and oxidative stability of sesame oil and can be applied in the oil industry.

## 1. Introduction

Sesame oil is recognized as a highly nutritional and healthy vegetable oil, which is rich in unsaturated fatty acids, such as oleic acid, linoleic acid, arachidonic acid, etc. It is very beneficial for lowering cholesterol, softening blood vessels, and preventing and treating diseases caused by vascular sclerosis [1]. Meanwhile, the oil also contains sesamol, sesamin, and vitamin E, which are beneficial to human health [2]. Generally, in order to improve the flavor and yield of the oil, sesame is usually roasted at a high temperature (180–220 °C) before pressing [3]. However, some nutrition substances, such as vitamin E, sesamol, and sesamin, are easily decomposed at a high temperature, and some carcinogens, such as benzopyrene and dioxin, are produced [4]. In order to ensure the oil’s safety and nutrition, decreasing the roasting temperature seems to be necessary, but the resulting oil with little prominent aroma would not be accepted [4,5]. The aromatic quality of sesame oil is mainly attributed to volatile compounds, including pyrazines, furans, thiazoles, pyrroles, alcohols, and aldehydes [3,6]. Specifically, methanethiol, dimethyl disulfide, methyl-pyrazine, 2-ethyl-6-methyl-pyrazine, 3-ethyl-2, 5-dimethyl-pyrazine, 2-furan methanol, etc., play important roles in the flavor of sesame oil [7]. Meanwhile, as reported, pyrazine and other substances might be the main ingredients of sesame oil essence [8]. Recently, cheap chemical synthetic substances were found to be illegally added into the high-temperature-treated sesame oils to create high-flavor sesame oil, which would become a problem for public health [9]. Therefore, it is necessary to develop a safe and effective flavor enhancement pathway for sesame oil.

At present, the Maillard reaction (MR) is the main research direction of food flavor enhancement. The reaction involves a series of interactions between components containing carbonyls (reducing sugars) and substrates containing amino components (proteins, polypeptides, and amino acids), resulting in many flavor components, including furans, pyrazine, pyrrole, thiophenes, thiazoles, and other heterocyclic compounds [10,11]. These flavor components can endow foods with multiple flavors [12]. Sesame meal is the main byproduct in sesame oil production that is rich in protein and starch. Nowadays, the meal is mainly used as fertilizers and feeds, resulting in a waste of resources. It could be used as a good source of substrates for the MR. Recently, there were many studies on the MR of sesame meal polypeptide systems combined with the addition of reducing sugars and amino acids, which could enhance the product flavor. Qin et al. [13] found that the oxidative stability of cold-pressed sesame oil could be improved by the Maillard products of peptides and reducing sugars in sesame meal. Hu et al. [14] improved the flavor and physicochemical properties of these salts using Maillard flavor products from sesame meal peptides. Shen et al. [15] used different sulfur-containing substances and defatted sesame meal to produce Maillard reaction products with high flavor acceptance, which could be used to improve the flavor of specific foods. However, there is no information about only using sesame meal to produce an MR and further improve the flavor of sesame oil.

The purpose of the present work was to improve the aroma quality of sesame oil by Maillard reaction products (MRPs) derived only from sesame meal without exogenous addition, and this method attempted to solve the problem of the insufficient aroma of sesame oil caused by lower-temperature (160 °C) roasting. Firstly, the sesame seeds were roasted and then pressed to obtain sesame crude oil and meal. Secondly, the meal was hydrolyzed with protease to obtain a polypeptide-hydrolyzed liquid, which was subsequently treated differently with (or without) glucoamylase. Afterwards, the enzymatic hydrolysate obtained was subjected to produce MRPs. Finally, the protease hydrolysate, protease-glucoamylase hydrolysate, and their MRPs were further blended with the crude oil to obtain the resulting oils. The changes in the quality of the resulting oils and crude oil were preliminarily compared. The colors of the oils were comparatively evaluated with a colorimeter. Their aromas were also evaluated by sensory analysis combined with an electronic nose, and their pivotal volatile components were analyzed by solid-phase microextraction–gas chromatography–mass spectrometry (SPME-GC-MS) and odor activity value (OAV). Finally, during accelerated oxidation, the oxidation stability of all the oils was preliminarily analyzed in terms of peroxide value and acid value. The research provides a feasible method to improve the fragrance of sesame oil without high-temperature (180–220 °C) roasting, which is expected to increase the value of the sesame meal and improve the oil quality.

## 2. Results and Discussion

### 2.1. Effect of Glucoamylase on Sugar Content in Hydrolysate

Sugars, especially reducing sugars, are an important reaction substrate of the MR. In the reaction, reducing sugars and amino components (amino, polypeptide, and protein) can be converted into Maillard products under heating conditions [16]. A higher amount of sugar would favor the production of Maillard products. In order to increase total sugars and reducing sugars, the meal was primarily hydrolyzed by glucoamylase. The sugar content changes are shown in Table 1. Clearly, the total sugar content of the hydrolysate increased from 10.23 to 10.58 mg/100 mL after treatment with glucoamylase as the starches were hydrolyzed into oligosaccharides, disaccharides, and monosaccharides under hydrolysis of glucoamylase. Moreover, the resulting oligosaccharides were further hydrolyzed to produce monosaccharides and disaccharides. The reducing sugar content in PH was 6.51 mg/100 mL, while GPH increased to 9.07 mg/100 mL. After glucoamylase treatment, the reducing sugars also increased, meaning that the starches and oligosaccharides containing α-1,4- and α-1,6 glycosidic bonds were hydrolyzed into monosaccharides and reducing disaccharides in the meal.

### 2.2. Changes in Sugars and Free Amino Acids after the Maillard Reaction

Sugars and amino acids are the main substrates of the MR, and their changes usually represent the degree and direction of the MR [17]. The content of sugars after the MR is shown in Table 1. After the MR, the total sugar content of PH and GPH was reduced to 8.04 and 8.18 mg/g, respectively. The consumption ratios of the total sugars were 20.04 and 24.01%, respectively. Clearly, the total sugar consumption of GPH was greater than that of PH. Additionally, reducing sugars had a similar tendency. The consumption ratio of reducing sugars in GPH (34.18%) was higher than that of PH (27.19%). This indicated that more sugars from the hydrolysate treated with glucoamylase participated in the MR. Therefore, after treatment with MR, more Maillard products were produced in the meal with glucoamylase hydrolysate.

The amino acids in the meal hydrolysate are shown in Table 2. After the MR, the total amount of free amino acids in PH was reduced. For example, the total amount of free amino acids in PHM was 473.92 μg/g, which was about 10% lower than PH (530.61 μg/g). The total amount of free amino acids in GPHM (462.23 μg/g) was about 13% lower than that of GPH (531.43 μg/g). Obviously, the free amino acids significantly (*p* < 0.05) decreased after the meal was treated with both glucoamylase and MR. Coincidentally, the decrease in amino acids was consistent with the change in reducing sugars and total sugars in Table 1, as they interacted with each other and were converted into Maillard products. Furthermore, different kinds of amino acids showed different downward trends. For example, the cysteine content was 0.35 μg/g in GPH and not detected in GPHM. The reduction in these amino acids might be due to the various MR degrees. Zhu et al. [18] found that histidine was one of the main sources of pyrazine in a Maillard reaction. The histidine content (45.39 μg/g) in GPHM was significantly (*p* < 0.05) lower than that of GPH (77.37 μg/g), implying that more histidine in the hydrolysate treated with glucoamylase was involved in the MR. This might be the main reason for the increasing pyrazine volatile components in the resulting oils [18,19].

### 2.3. Color Changes in the Sesame Oils

Color is another important indicator for assessing the quality of oils [4]. The color values (a*, b*, and L*) and total color difference (ΔE*) of the oils are displayed in Figure 1A,B. The values of a* and b* of the various oils are exhibited in Figure 1A. After adding the MRPs of the meal hydrolysate with (or without) glucoamylase, a* and b* values of the oils changed nonsignificantly (*p* > 0.05), such as the a* value varying from 0.31 (C-SO) to 0.36 (GPHM-SO) and the b* value decreasing from 1.07 (C-SO) to -1.61 (PHM-SO). Moreover, after the oil was treated with the MRPs of the meal hydrolysate with glucoamylase, the a* value increased significantly (*p* < 0.05), which represented an increase in oil redness. In Figure 1C, it can be intuitively observed that the processed oil (GPHM-SO) has the reddest color. The increase in a* and b* values might indirectly reflect that more brown substances were produced in the resulting oils after being treated with the meal hydrolysate and MRPs.

As shown in Figure 1B, the L* (histogram) values of the PHM-SO and GPHM-SO were 54.60 and 54.58, respectively. Compared with other oils, the L* values of both oils significantly (*p* < 0.05) decreased, indicating that the color of the oils treated with MRPs was darker than that of the other oils. The increase in natural pigment components, such as sesamol, sesamin, and sesamolin, would make the color darker. Moreover, colored components such as flavonoids and melanoidins were also produced in the MR of the meal hydrolysate and caused the colors of oils to deepen. The ΔE* values of the oils are shown in Figure 1B (dashed line), and this value of the oil (GPHM-SO) treated with the Maillard products of the meal hydrolysate with glucoamylase hydrolysate was higher than that of the other oils, as more the pigments and Maillard products entered the oil after blending.

### 2.4. Sensory Evaluation

The sensory properties of the oils treated with different meal products are listed in Table 3. Compared with crude oil, after being treated with PH, GPH, PHM, and GPHM, the flavor attributes of the oils were improved. For instance, the crude oil treated with GPHM was not only enhanced in the toasted sesame aroma, caramel aroma, and roasted nut aroma but also lower in the off-flavor. The overall satisfaction with C-SO, PH-SO, GPH-SO, PHM-SO, and GPHM-SO was in the respective order of 5.65 < 5.85 < 5.93 < 6.21 < 7.08. Obviously, in all the oils, after being treated with GPHM, the overall satisfaction with the oil (GPHM-SO) was increased significantly (*p* < 0.05). Therefore, the Maillard products of the meal hydrolyzed by glucoamylase had a positive effect on the oil flavor, which could improve the sensory quality of the oil [20]. After the meal was hydrolyzed by glucoamylases and proteases, the reducing sugars and free amino acids increased, which were converted by the Maillard reaction into more flavor products, enhancing the aroma of the sesame oil.

### 2.5. Electronic Nose Analysis of the Oils

The distinction of volatiles between different sesame oils was analyzed by the electronic nose technology [6]. The PEN-3 electronic nose was equipped with 10 sensors, and each, namely R1, R2, R3, R4, R5, R6, R7, R8, R9, and R10, could sensitively distinguish aromatic, nitrogen oxide, ammonia–aromatic, hydrogen, alkane–aromatic, broad-methane, sulfide, broad-alcohol, organic sulfur–aromatic, and broad-alkane compounds, respectively. The electronic nose mainly reflects the gas content by observing the intensity of the different sensors. A high signal intensity of a sensor indicates a high content of the corresponding substances [21]. The aroma profiles of all the oils were analyzed by the electronic nose, and the sensor data are presented in Figure 2. Clearly, the signal strengths of R2 (nitrogen oxide), R6 (methane), R7 (organic sulfides), and R9 (sulfides) sensors were higher than that of the other sensors. However, the signal changes in the other sensors were not obvious. This indicated that the main aromatic compounds in sesame oil were nitrogen oxides and sulfides, which is consistent with the report of Park et al. [22]. Interestingly, signal intensities of the R2, R7, and R9 sensors of the treated oils were higher than those of the crude oil. In particular, R2 (7.91), R7 (16.38), and R9 (10.41) of the oil treated with GPHM (GPHM-SO) were the highest in all the oils, indicating that the contents of nitrogen oxides and sulfur compounds increased maximally in the oil treated with GPHM. In short, the nitrogen oxides and sulfur compound species increased after the oil was mixed with the MRPs of the meal hydrolysate with glucoamylase.

In order to directly observe the volatile compound difference among sesame oils, principal component analysis (PCA) was performed to explore the principal components that contribute the most to the overall data by reducing the dimensionality of multiple sensor data from the electronic nose [23]. The PCA results of different oils are presented in Figure 2. The first and second principal components (PCs) accounted for 71.6 and 23.7% of the total contribution rates, respectively. The sum of the first two PCs represented 95.3% of the total sample variance, presenting sufficient information for explaining differences in the aromas of sesame oils. The spatial distribution of the five samples did not overlap, indicating that the oil odors could be distinguished by the electronic nose [24]. Meanwhile, according to the results of the oils’ quadrant distribution, the oil samples were divided into three groups. The principal components of HP-SO, GPH-SO, and PHM-SO were distributed in the first quadrant, C-SO in the second quadrant, and GPHM-SO in the third quadrant. The distance between the three groups was scattered, indicating that their principal components were significantly different. The division of these three parts was similar to the overall aroma score ranking of the oils. In short, from electronic nose analysis, the meal hydrolysate and treatment with its MRPs could significantly change the types and contents of the volatile compounds of the resulting oils.

### 2.6. Volatile Compound Analysis

#### 2.6.1. Volatile Compound Changes in Treated Oils

The volatiles of sesame oil after different treatments were further identified by SME-GC-MS and compared with common sesame oils on the sale (S-SO). As illuminated in Table 4, there were 49 volatile compounds including 11 pyrazines, 7 furans, 5 sulfur compounds, 5 pyrroles, 3 pyridines, 5 aldehydes, 5 ketones, 5 phenols, and 3 alcohols, which were identified and quantified in all the oils. The treated oil sample was similar to market-available sesame oil in the type of volatile compounds. Usually, volatile compounds such as nitrogen heterocyclic (pyrazines and pyridines), oxygen heterocyclic (furans), and sulfur compounds (thiols, thiophenes, and thiazoles) have the greatest impact on sesame oil aroma [25].

The nitrogen-containing heterocycles mainly existed in the form of pyrazines, pyridines, and pyrroles. Pyrazine compounds in sesame oil were the main contributors to the aroma and were produced usually in an MR [6,19]. Generally, an MR begins with the condensation between carbonyls and amino groups to form glycoslamine compounds, which then are converted into Amadori products and then subjected to dehydration and enolization, followed by the deamination reaction, in which dicarbonyl compounds and other substances are finally formed. Further, after Strake degradation, amino groups are combined with dicarbonyl compounds and condensed into pyrazines [12,26]. The total pyrazine content of the different oils is depicted in Appendix A. The total pyrazine content in C-SO, HP-SO, GPH-SO, PHM-SO, GPHM-SO, and S-SO was 65.07, 86.20, 88.61, 104.38, 114.48, and 97.18 μg/mL, respectively. Clearly, pyrazine content in the oil treated with the hydrolysate was slightly higher than that in crude oil. This might be due to the Strake degradation of some amino substances and dicarbonyl compounds during the mixing process to produce pyrazine, resulting in an increase in the pyrazine content of the oil. In addition, MRP-treated oils (PHM-SO and GPHM-SO) had higher levels of pyrazine than other oils, and the pyrazine content in GPHM-SO was especially and significantly (*p* < 0.05) increased. This might be due to the increase in carbonyl substances (monosaccharides and oligosaccharides) in the hydrolysate treated with glucoamylase, which led to the increase in Amadori products. Since the degradation process of Strake was promoted, the pyrazine content in the meal increased. When the meal was fully blended with the oil, more pyrazines inevitably entered into the oil. Moreover, the components and contents of the flavor substances in the different oils are listed in Table 4. Compared with C-SO and S-SO, after being treated with the MRPs of the meal hydrolysate with glucoamylase, pyrazines in the oil increased differently. For example, the content of methyl-pyrazine (chocolate odor) was 13.48, 28.28, and 32.38 μg/mL in C-SO, S-SO, and GPHM-SO, respectively. Similarly, 2,5-dimethylpyrazine (coffee nut odor) content was 3.44, 16.56, and 18.23 μg/mL in the three oils, respectively. Methyl-pyrazine and 2,5-dimethylpyrazine were commonly found in some common sesame products, such as sesame oil and roast sesame, and were closely related to the roasted sesame aroma [20]. The content of nitrogen-containing heterocycles and some special pyrazines increased after the oil was treated with GPHM, which might relate to the improvement in the overall flavor of the resulting oil. The changes in pyrazines were consistent with the sensory analysis and previous results [10,11].

The oxygen-containing heterocycles, which contributed to the flavor of sesame oil, mainly existed in the form of furans. Furthermore, sulfur-free furans were very important flavor substances and made an important contribution to the caramel and nutty aroma of sesame oil [27]. Sulfur-free furans can be produced by amino acids in the MR. Amino acids can be degraded to produce acetaldehyde or glyceraldehyde, which can be further cyclized, dehydrated, and finally condensed to form furans. Meanwhile, furans can also be generated under the conditions of dehydration, cyclization, polymerization, and fragmentation. As shown in Appendix A, the total content of furans was 9.31, 10.52, 10.75, 12.59, 13.95, and 11.61 μg/mL, respectively, in C-SO, HP-SO, GPH-SO, PHM-SO, GPHM-SO, and S-SO. Obviously, the total furans found in the oils treated with GPHM increased significantly (*p* < 0.05), as there were more monosaccharides and oligosaccharides in sesame meal after being treated with glucoamylase. The components and contents of flavor substances in the different oils are listed in Table 4. Components such as 5-methyl-2-furfural and 2-furan-methanol were detected in all oils, while the highest contents of 7.09 and 3.42 μg/mL, respectively, were found in GPHM-SO. Moreover, 5-methyl-2-furfuraldehyde and 2-furan-methanol might be the common furan volatile components in sesame oil and usually have a sweet fragrance. These furans might be the key to increasing the caramel aroma of the oil, which is consistent with the results of Yin et al. [19]. The increase in these specific aroma substances was one reason for the overall flavor enhancement of the resulting oil.

Sulfur compounds have a low odor threshold and play an important role in the aroma of different foods, and they also play key regulatory roles in the aroma composition of sesame oil [6]. The formation of sulfides in the MR is similar to that of pyrazines, which are formed by condensation of sulfur-containing amino acids followed by Strake degradation. The total content of sulfur compounds in the oils is shown in Appendix A. After the addition of the meal hydrolysate and their MRPs in the crude oil, the composition of sulfur compounds increased significantly (*p* < 0.05), especially in the oil treated with GPHM, and GPHM-SO (3.58 μg/mL) had the highest sulfur content because more sulfur compounds entered into the oil during the mixing process. In particular, as listed in Table 4, dimethyl trisulfide (fresh onion) was identified in all the oils, and it is regarded as an important contributor to the sesame flavor. The dimethyl trisulfide content in GPHM-SO was 1.14 μg/mL, which was slightly higher than in other oil samples. Compared with the other oils, the contents of 4-methylthiazole (a pleasant nutty flavor) and 2,4-dimethylthiazole (coffee flavor) in GPHM-SO were 0.94 and 0.72 μg/mL, respectively, which also were the highest. These sulfides had a great influence on the improvement of the oil flavor. The increased content of the sulfides might be one of the reasons for the increased aroma abundance of the oils, which is consistent with the report by Park et al. [22].

Aldehydes, ketones, phenols, and alcohols generally do not make a significant contribution to the organoleptic properties of sesame oil due to their high odor thresholds. However, these substances can not only act as important intermediates for the formation of heterocyclic compounds in the MR but also play coordinating, complementary, and enriching roles in the oil’s overall flavor [26,28,29]. As shown in Appendix A, the total contents of aldehydes in PH-SO and GPH-SO were 8.25 and 8.44 μg/mL, respectively, and were significantly (*p* < 0.05) lower than C-SO (11.07 μg/mL). This might be due to the condensation reaction between the carbonyl aldehydes of sesame oil and the amino substances in the enzymatic hydrolysis during the mixing process, resulting in the reduction in aldehydes [30]. Compared with the others, the total amount of aldehydes (11.44 μg/mL) increased after the oil was treated with GPHM, which was related to the high content of aldehydes in the MRPs resulting in more aldehydes entering sesame oil. For the content of different aldehydes, hexanal (0.88 μg/mL) was only detected in GPHM-SO with a low aroma threshold and a fresh fruity aroma, while GPHM-SO had the highest content of benzaldehyde (2.40 μg/mL). Benzaldehyde from amino acid degradation and lipid oxidation was nutty and sweet and previously detected as an aroma-active compound in roasted sesame oil [6].

In addition, some phenols and aldehydes also had positive effects on the oil aroma. The content of phenolic substance 2-methoxyphenol (smoky fragrance) in GPHM-SO (12.81 μg/mL) was higher than that in C-SO (8.79 μg/mL). In all the oils, as an alcohol, 1-octen-3-ol, which had a grassy and earthy aroma, was the highest (1.37 μg/mL) in content in GPHM-SO. In brief, the addition of sesame meal hydrolysate and its MRPs into the oil could greatly increase the amount of carbonyl compounds and specific flavor substances, especially for the GPHM-SO group.

#### 2.6.2. Aroma Activity Value Analysis

Although there were many volatile compounds in sesame oil, only a few of them contributed significantly to the overall flavor. Generally, the contribution of the volatile compounds to sesame oil aroma depends on their content and odor threshold. Aroma usually is expressed in terms of odor activity values (OVAs) [19]. Table 5 lists the odor activity values of the major volatile compounds in the oils (mainly listing substances with OAV ≥ 1). Clearly, there were 11 major volatile compounds that were common to all the oils, namely 2,5-dimethylpyrazine, 2,6-dimethylpyrazine, 2,3-dimethylpyrazine, 2-ethyl-5-methylpyrazine, trimethyl-pyrazine, 3-ethyl-2,5-dimethylpyrazine, 2-pentylfuran, dimethyl-trisulfide, benzaldehyde, 2-methoxyphenol, and 1-octen-3-ol. These substances could be assumed as the key aromatic compounds that comprised the oil aroma, which is fairly consistent with the results reported by Jia et al. [6]. As shown in Table 5, the odor activity values of these substances in the different treated oils were improved to various degrees. In all the oils, the odor activity values of GPHM-SO increased most significantly. The activity value of 2,3 dimethyl-pyrazine with a cocoa flavor in GPHM-SO (31.38) was higher than other oils. The aroma activity values of 2,5-dimethylpyrazine and 2,6-dimethylpyrazine in GPHM-SO were 18.23 and 15.11, respectively, which were the highest of all the oils. The improvement of the aroma activity values played a key role in the aroma enhancement of the oils. Compared with crude oil, GPHM-SO also had three additional characteristic aromatic compounds, which were methylpyrazine (1.20), 4-methylthiazole (17.09), and hexanal (11.00). These substances provided the possibility of enriching the aroma of sesame oil, and these changes were the main reasons for the highest comprehensive sensory score for the oils treated with the MRPs of the meal hydrolysate with glucoamylase.

### 2.7. Oxidative Stability of Sesame Oil

The oxidative stability of the different treated oils was also preliminarily evaluated in the accelerated oxidation experiment. The peroxide value and acid value of the oils during the 25 days of the oven experiment are shown in Figure 3. As shown in Figure 3A, with the increasing storage time, the peroxide value of all the oils increased gradually. The peroxide value of the crude oil increased most notably from 0.09 to 4.92 mmol/kg, while that of the oil treated with GPHM increased only from 0.11 to 3.03 mmol/kg. The peroxide value is one of the important indices in the initial oxidation stage of oils as a higher value indicates serious oxidation of the oil [31]. The oxidative stability of all the treated oils was improved compared with that of the crude oil. Moreover, after being treated with GPHM, the stability of oils was improved significantly (*p* < 0.05). This might be related to melanoidin and a large number of volatile heterocyclic compounds after the MR. After these components enter the oil, they inhibit lipid oxidation by inhibiting the formation of oil peroxide and reducing the decomposition of hydroperoxide into free radicals [32]. As shown in Figure 3B, with the increasing storage time, the acid value changed little, ranging from 0.20 to 0.35 mg KOH/kg. The acid value is an indicator of the content of free fatty acids produced by fat hydrolysis, and it is one of the indicators for evaluating the later stage of oil oxidation [31]. The acid value of all the oils changed conspicuously during the storage, indicating that the hydrolysis reaction of all oils hardly occurred during the accelerated oxidation period, which is consistent with previously reported results [32].

Taking peroxide value as the evaluation object, the relationship between storage time (x) and peroxide value (y) was fitted by a logistic function (the fitting curve is shown in Appendix A). As shown in Table 6, the R^2^ values of all oil sample fitting equations were above 0.95, implying great fitting results. The fitting method could be used to predict the change in peroxide value in the process of accelerated oxidation. The forecasted time was calculated by substituting the lowest standard peroxide value (Chinese national standard GB/T 8233-2018) of first-order sesame oil into the fitting equation. The predicted storage times of C-SO, HP-SO, GPH-SO, PHM-SO, and GPHM-SO were 31, 33, 33, 53, and 60 days, respectively. Clearly, the oil treated with GPHM had the longest storage time and the best oxidation stability of all oils. It could be speculated that the MRPs of the enzymatic hydrolysis of sesame meal endowed the oil with more natural antioxidant ingredients, which effectively prevented oil oxidation. However, the effect on the later stage of lipid oxidation remains to be studied.

## 3. Materials and Methods

### 3.1. Chemicals and Materials

Ganzhi No. 10 white sesame was provided by Dayuan Oil Co., Ltd. (Suzhou, China). Sesame oil was provided by Yushan Oil Co., Ltd. (Chuzhou, China). Glucoamylase (100 U/mg), neutral protease (100 U/mg), and flavor protease (20 U/mg) were purchased from Shanghai Yuanye Biotech. Co., Ltd. (Shanghai, China). 1,2-dichlorobenzene (chromatographic grade) was purchased from McLean Bioch Tech. Co., Ltd. (Shanghai, China). All other solvents and reagents with analytical-grade purity were provided by Jiangsu Jincheng Reagent Co., Ltd. (Suzhou, China).

### 3.2. Sample Preparation

#### 3.2.1. Preparation of Sesame Oil and Sesame Meal

The pressing process was carried out according to the method of Ji [3]. The sesame seeds (2.00 kg) were roasted in an electric roaster (PEO-1107, Kerong Electric Ltd., Foshan, China) at 160 °C for 10 min. Then, the roasted seeds were pressed with a screw press machine (X8S, Xiangju Intelligent Ltd., Shenzhen, China) to obtain the crude oil and sesame meal. The meal was dried in a blast drying oven (DHG-9070A, Jinghong Experimental Equipment Ltd., Shanghai, China) at 60 °C until the moisture content was <5%.

#### 3.2.2. Preparation of Sesame Meal Hydrolysate

The meal hydrolysate was prepared using the method of Qin [13] with appropriate modifications. The meal was crushed with a high-speed grinder (800Y, Boou Hardware Manufacturing Ltd., Ningbo, China) and further filtered through a 100-mesh sieve to obtain the powder (protein content 35.23% and starch content 4.56%, which were measured according to GB 5009.5-2016 and GB5009.9-2016 National Standards of China, respectively). The powder was mixed with ultrapure water in a 1:10 (w/v) ratio and preheated at 90 °C for 20 min. The mixture was adjusted to pH 4.5 with 6 M citric acid after cooling, and the mixture was hydrolyzed at 55 °C for 3.5 h with glucoamylase (300 U/g, calculated as substrate starch content). Meanwhile, the mixture without glucoamylase was set as control. The above hydrolysate was adjusted to pH 7.5 with 6 M Na_2_CO_3_ and then cohydrolyzed at 55 °C for 4.0 h with 5000 U/g neutral protease and 800 U/g flavor protease (calculated as substrate protease content). The sesame meal protein hydrolysate without glucoamylase was denoted as PH. The sesame meal hydrolysate, to which glucoamylase and protease were added, was recorded as GPH. Finally, the enzymes were inactivated at 90 °C for 15 min. Then, the hydrolysate was centrifuged at 8000 rpm for 20 min using a high-speed centrifuge (TGL16M, Yancheng KaiT Experimental Instrument Co., Ltd., Yancheng, China). Finally, the supernatant was collected and stored at −20 °C for later use.

#### 3.2.3. Preparation of Maillard Reaction Products

The Maillard products were prepared according to the method of Zhang et al. [17] with some modifications. The pH of the meal hydrolysates (PH and GPH) was adjusted to 7.5 with Na_2_CO_3_ (6 mol/L), and the hydrolysates were then transferred into a 100 mL round-bottomed flask. The hydrolysates were transferred into an oil bath and heated in a thermostatic oil bath (DF-101Z, Shanghai LICHEN-BX Instrument Technology Co., Ltd., Shanghai, China) at 120 °C for 1 h with magnetic stirring at 800 r/min. At the end of the reaction, the liquid Maillard products of PH and GPH were denoted as PHM and GPHM, respectively.

#### 3.2.4. Preparation of Oils Using MRPs

The main preparation flowchart for refining sesame oil is shown in Figure 4. Specifically, the sesame meal protease hydrolysate, protease-glucoamylase hydrolysate, and their MRPs (PH, GPH, PHM, and GPHM) were mixed with the crude oil in a ratio of 1:8 (*v*/*v*). The mixture was stirred for 6 h (400 r/min) at room temperature (25 ℃), and then the water in the oil was dried with anhydrous sodium sulfate. After centrifugation at 6000 rpm for 5 min, the oils obtained were recorded as PH-SO, GPH-SO, PHM-SO, and GPHM-SO, while the crude oil was recorded as C-SO. Then, all the oils were sealed and stored in the dark for later use.

### 3.3. Analysis Methods

#### 3.3.1. Determination of Reducing Sugar and Total Sugar Content

Reducing sugar and total sugar were detected using the method of Song et al. [33] with appropriate modifications. The 5 mL aliquots of the samples were dissolved in 25 mL of deionized water in the reducing sugar assay. Secondly, the pretreated sample (2 mL) was mixed with DNS (2 mL) reagent (with deionized water as the control) and reacted at 90 °C for 6 min. After the mixture was cooled to room temperature, an ultraviolet–visible spectrophotometer (UV-4802, Unico Shanghai Instrument Co., Ltd. Shanghai, China) was used to determine the absorbance of the mixture at 540 nm. Finally, the contents of reducing sugar and total sugar were determined according to the standard glucose curve (y = 8.9174X-0.0118, R^2^ = 0.991, Appendix A).

For determination of total sugar, the sample (5 mL) was mixed with HCl (10 mL, 6 mol/L) in a boiling water bath for 10 min, the pH was adjusted to 7.0 with NaOH (6 mol/L) after the mixture was cooled to room temperature, and then the volume was fixed to 25 mL with deionized water in the total sugar assay. The pretreated sample (2 mL) was mixed with DNS, and the subsequent procedure was consistent with that of reducing sugar determination.

In order to better reflect the consumption degree of the sugars in the hydrolysate after MR, the consumption rate of the sugars after the reaction was determined. Equation (1) was used to calculate the consumption rate of sugars as follows:(1)$$\mathrm{Consumption}\;\mathrm{rate}\;(\%)=\frac{\mathrm{Sugar}\;\mathrm{content}\;\mathrm{before}\;\mathrm{reaction}-\mathrm{Sugar}\;\mathrm{content}\;\mathrm{after}\;\mathrm{reaction}}{\mathrm{Sugar}\;\mathrm{content}\;\mathrm{before}\;\mathrm{reaction}}\times 100\%$$

#### 3.3.2. Determination of Free Amino Acids (FAAs)

The content of free amino acids (FAAs) in the meal hydrolysates before and after the MR was detected using a fully automated amino acid analyzer (L-8900, Hitachi Ltd., Tokyo, Japan) [17]. The sample (after freeze-drying 0.04 g) was dissolved in 10% trichloroacetic acid (6 mL) and placed at 4 °C overnight. After centrifuging at 10,000 rpm for 20 min, the supernatants were filtered with a 0.22 μm membrane. Then, the filtrates were quantitatively analyzed based on the retention time and peak area of the fingerprints of the 17 amino acid standard mixtures (016–08641, Wako Pure Chemical Industries, Ltd., Tokyo, Japan).

#### 3.3.3. Determination of Color

The oil colors were measured with a portable colorimeter (NR200, GuanLian Industrial Ltd., Shenzhen, China). The colorimeter was calibrated with a white surface calibration plate prior to analysis. Then, the color-attributed values of L* (brightness (+) or luminosity (−)), a* (redness (+) or greenness (−)), and b* (yellowness (+) or blueness value (−)) were measured according to the Commission International Eclairage (CIE) Lab color scale. The total color difference (ΔE) was then calculated according to Equation (2), which is expressed below.
(2)$$\Delta E=\sqrt{\Delta L^{*2}+\Delta a^{*2}+\Delta b^{*2}\;\;}$$

#### 3.3.4. Descriptive Sensory Analysis of Sesame Oil

For the sensory analysis of the oils, 10 well-trained personnel (5 male and 5 female) were selected from the Scientific Sensory Evaluation Laboratory of Hefei University of Technology to perform the descriptive sensory analysis. Before the sensory evaluation, an agreement on descriptive sensory evaluation was established through a full discussion on specific indicators of toasted sesame aroma, caramel aroma, aroma persistence, off-flavor, and overall satisfaction. The roasted sesame aroma of sesame oil was evaluated using the aroma of roasted sesame seeds as a criterion. The caramel flavor of the oil was evaluated using the aroma of roasted butter scotch. The aroma persistence was evaluated according to the first-class sesame oil sold on the market, and the off-flavor was evaluated according to burnt sesame seeds. Then, overall satisfaction was based on the previous four sensory evaluations. The specific contents of the sensory evaluations are listed in Appendix A. The sensory evaluation was performed at room temperature (25 °C) and completed within one day. The sensory evaluation of each sample was repeated three times.

#### 3.3.5. Electronic Nose Analysis of Sesame Oil

The aroma of the oils was analyzed by an electronic nose (PEN3, AIRSENSE Analytics GmbH, Schwerin, Germany). Three milliliters of each oil was added in a beaker (50 mL) with 3 layers of plastic wrap and equilibrated in a 50 °C water bath for 20 min. During the measurement, the equipment was flushed with compressed air for 60 s while the flow rate of the compressed air was 400 mL/min. Then, the samples were automatically detected. The detection interval was 1 s, while the detection time was 100 s. The experiment data of 57–60 s were selected for subsequent analysis.

#### 3.3.6. Volatile Compound Analysis by SPME-GC-MS

The determination of volatile compounds in the oils was adapted from the method of Yin et al. [19]. In this measurement, 10 μL of 1,2-dichloroobenzene (50 μg/mL in methanol) used as an internal standard was added into the oils (2 mL). Each sample that contained the internal standard was placed in a 15 mL headspace bottle and sealed with a PTFE isolation cap. Then, the volatile substances were adsorbed by the DVB-CAR-PDMS extraction head (Supelco Co., Bellefonte, PA, USA), which was aged for 1 h at 250 °C the first time. The aged extraction head was inserted into the sample bottles in a water bath at 60 °C for 20 min and further adsorbed for 30 min. The adsorbed extraction head was inserted into the GC-MS (QP2010SE, GK/J-0950, Shimadzu, Excellence in Science, Inc, Tokyo, Japan) instrument’s inlet pyrolysis adsorption for 6 min before further analysis. The gas chromatography column was HP-5MS (30 m × 0.25 mm × 0.25 μm). The column temperature program was kept at 60 °C for 2 min and then at 60–240 °C at a rate of 5 °C/min maintained for 10 min. The inlet temperature was maintained at 240 °C. The carrier gas was high-purity helium at a flow rate of 1.2 mL/min, and the distillation extraction split ratio was 10:1 (*v*/*v*). The mass detector was used with electron impact at MS 70 eV. The ion source temperature was fixed at 230 °C while the spectrum was obtained in the mass range of 40–450 m/z. The volatile components were identified based on computer matching with the NIST Library mass spectrometry library.

The approximate quantities of the volatile compounds were estimated by the following Equation (3), with 1,2-dichlorobenzene as the internal standard and a calibration factor of 1.
(3)$$C_{v}=\frac{S_{v}}{S_{i}}C_{i}$$
where S_v_ and S_i_ represent the peak areas of volatile compounds and the internal standard, respectively, while C_v_ and C_i_ represent the concentrations of volatile compounds and the internal standard, respectively.

#### 3.3.7. Calculation of Odor Activity Value

The OAV was used to evaluate the contribution of the volatile compounds to the oil fragrance, which represented the contribution degree of a single odor to the overall aroma. A volatile substance OAV ≥ 1 reflected that the substance had an important contribution to the overall aroma of the oils [34]. The odor activity value (OAV) was calculated using the method of Zhou et al. [35].
(4)$$\mathrm{OAV}=\frac{C_{i}}{{\mathrm{OT}}_{i}}$$
where C_i_ is the relative concentration of the compound in the oil, and OT_i_ is the odor threshold of the compound in the oil.

#### 3.3.8. Oxidative Stability Analysis of Sesame Oil

The accelerated oxidation experiment of the oils was performed referring to the experimental method of Pereira [36] with appropriate modifications. Ten grams of each oil was put into a 25 mL glass vial, which was then sealed with a Teflon-coated rubber septum and aluminum cap. The vials were placed in an electrothermal incubator (DHP-48/95B, Tianjin Sedelis experimental and Analytical Instrument Manufacturing Co., Ltd., Shenzhen, China) at 60 ± 3 °C for 25 days to accelerate their oxidation. The oils were taken out every 5 days for analysis of peroxide value (POV) and acid value (AV). The samples were prepared in duplicate. The titration method (AOCS official method Cd 8b-90, 2009) of oils against iodine sodium thiosulfate solution was applied to assess the POV of the oils, and the AV was determined as described using the AOCS official method Ca 5a-40 (AOCS, 2009).

### 3.4. Statistical Analysis

The data were analyzed using SPSS 24.0 (SPSS Inc., Chicago, IL, USA) software. All data were calculated as mean ± standard deviation (SD, *n* = 3) and considered for significance at *p* < 0.05. Electronic nose data analysis was processed by the equipment’s own analysis software to perform principal component analysis (PCA) and established a radar fingerprint according to the sensor response value of the sample at 60 s. Other data processing was performed using Origin 2018 and Microsoft Office Excel 2010.

## 4. Conclusions

In this study, sesame meal hydrolysates were treated with protease and glucoamylase to obtain the amino acids and reducing sugars, then the Maillard products were firstly processed without exogenous additions, and the effects of PH, GPH, PHM, and GPHM on the aroma of low-temperature-roasted sesame oil were investigated. It was found that GPHM could significantly improve the aroma of oil with the highest contents of pyrazines, furans, and sulfur compounds in the oils and the highest odor activity value of the resulting oil. Furthermore, compared with the crude oil, three additional characteristic aroma compounds including methyl-pyrazine, 4-methylthiazole, and hexanal in GPHM-SO were found to have positive effects on aroma enhancement. Moreover, with the application of MRPs in sesame oil, GPHM-SO presented the best oxidation stability with the lowest peroxide value and acid value changes during the accelerated oxidation experiments, while the shelf life was predicted to be increased by 1.9 times compared with that of the crude oil by peroxide value fitting. This work proposes a new pathway for producing low-temperature-roasted sesame oil with better aroma and oxidation stability, and the sesame meal could be well utilized to improve the oil quality based on enzymatic hydrolysis and the Maillard reaction technique. In future studies, the composition of the interesting MRPs without exogenous amino acids and reducing sugars should be carefully investigated, while the MR conditions should be optimized for better application.

## Figures and Tables

**Figure 1 molecules-27-08857-f001:**
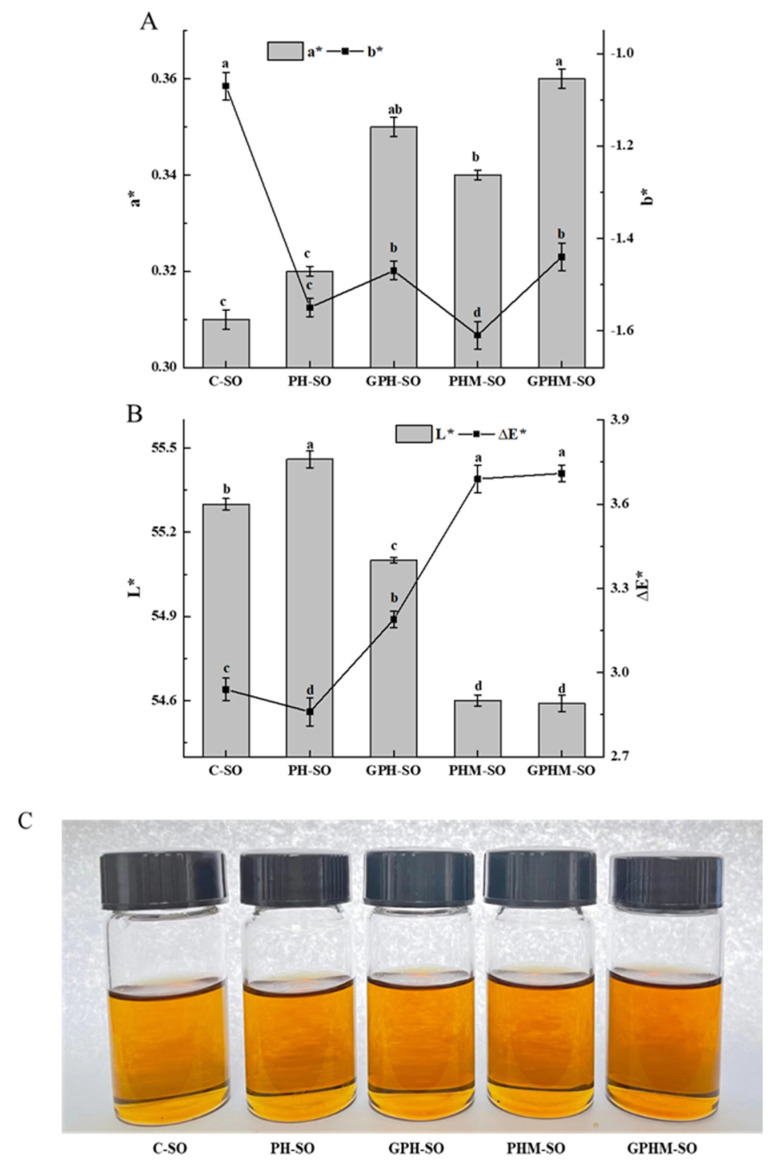
Color change in the oils with different treatments. (**A**) Redness or greenness (a*) and yellowness or blueness (b*) value of sesame oils, (**B**) brightness or luminosity (L*) and total color difference (ΔE*) of sesame oils, and (**C**) appearance of different processed sesame oils. Note: C-SO stands for the crude oil, PH-SO stands for the oil treated with hydrolysate without glucoamylase, GPH-SO stands for the oil treated with hydrolysate with glucoamylase, PHM-SO stands for the oil treated with MRPs of the hydrolysate without glucoamylase, and GPHM-SO stands for the oil treated with MRPs of the hydrolysate with glucoamylase. Values followed by different lowercase letters (a–d) mean statistically significant differences (*p* < 0.05) among different kinds of oils.

**Figure 2 molecules-27-08857-f002:**
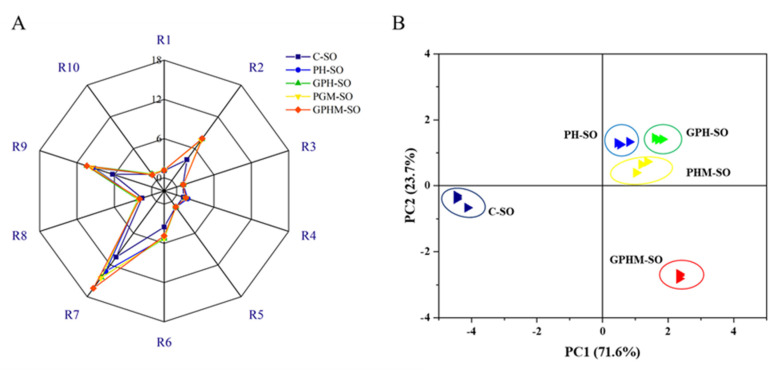
Electronic nose analysis of sesame oil with different treatments: (**A**) aromatic species analysis, (**B**) principal component analysis. Note: Sensors (R1, R2, R3, R4, R5, R6, R7, R8, R9, and R10) represented sensitivity for aromatic, nitrogen oxide, ammonia–aromatic, hydrogen, alkane–aromatic, broad-methane, sulfide, broad-alcohol, organic sulfur–aromatic, and broad-alkane compounds, respectively.

**Figure 3 molecules-27-08857-f003:**
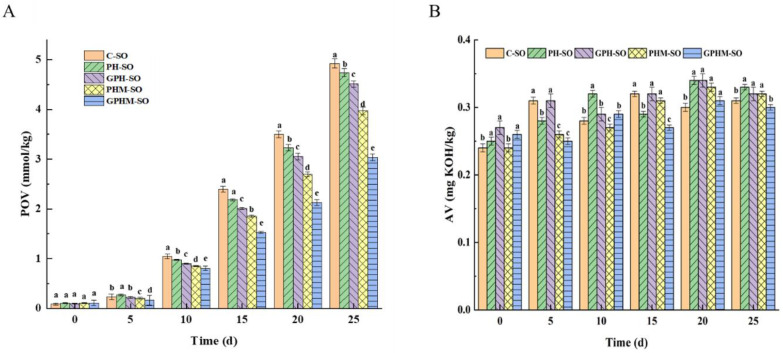
Changes in oxidative stability of the resulting oils during 60 ± 3 °C accelerated oxidation. (**A**) Peroxide value and (**B**) acid value. Note: Values followed by different lowercase letters (a–e) mean statistically significant differences (*p* < 0.05) among different kinds of oils.

**Figure 4 molecules-27-08857-f004:**
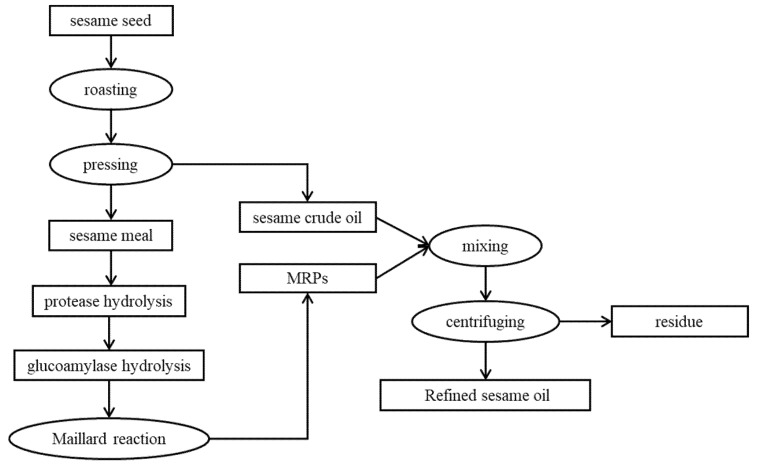
Preparation process of refining sesame oil.

**Table 1 molecules-27-08857-t001:** Changes in sugar content and sugar consumption ratio before and after MR in meal hydrolysate.

	No Glucoamylase Treatment			Glucoamylase Treatment		
Type of Sugar	Sugar Content (mg/g)		Consumption Rate (%)	Sugar Content (mg/g)		Consumption Rate (%)
	PH	PHM		GPH	GPHM	
Total sugar	10.23 ± 0.36	8.18 ± 0.34	20.04	10.58 ± 0.28	8.04 ± 0.25	24.01
Reducing sugar	6.51 ± 0.21	4.74 ± 0.18	27.19	9.07 ± 0.24	5.97 ± 0.22	34.18

Note: PH stands for the meal treated without glucoamylase, PHM stands for the MRPs of the hydrolysate without glucoamylase, GPH stands for the meal treated with glucoamylase, and GPHM stands for the MRPs of the hydrolysate with glucoamylase.

**Table 2 molecules-27-08857-t002:** Changes in free amino acids in the hydrolysate and the corresponding MPRs.

No.	Amino Acids	Free Amino Acids (µg/g)			
		PH	GPH	PHM	GPHM
1	Asp	19.48 ± 0.3 b	22.17 ± 0.59 a	14.41 ± 0.67 d	17.19 ± 0.35 c
2	Thr	9.11 ± 0.66 c	9.37 ± 0.75 c	12.72 ± 1.06 a	10.52 ± 1.01 b
3	Ser	2.85 ± 0.72 ab	3.62 ± 0.45 a	2.82 ± 0.27 b	3.38 ± 0.29 ab
4	Glu	4.37 ± 0.13 b	6.33 ± 0.30 a	4.68 ± 0.23 b	6.82 ± 0.34 a
5	Gly	10.31 ± 0.31 b	10.49 ± 1.10 a	9.89 ± 0.44 b	10.70 ± 0.75 a
6	Ala	51.32 ± 0.96 b	52.80 ± 2.33 a	47.85 ± 0.72 d	49.49 ± 0.92 c
7	Cys	0.41 ± 0.03 b	0.35 ± 0.09 b	-	-
8	Val	26.32 ± 1.24 a	28.43 ± 1.30 ab	23.17 ± 1.74 c	25.14 ± 1.69 bc
9	Met	69.56 ± 1.73 a	69.07 ± 1.44 a	64.15 ± 1.78 b	64.33 ± 1.89 b
10	Ile	123.27 ± 4.45 a	123.30 ± 3.64 a	116.25 ± 3.32 ab	114.82 ± 3.23 b
11	Leu	7.07 ± 0.78 b	10.33 ± 0.77 a	6.39 ± 0.71 b	9.62 ± 0.72 a
12	Tyr	91.77 ± 3.04 a	89.39 ± 3.40 a	83.49 ± 2.42 b	80.42 ± 1.83 b
13	Phe	10.30 ± 0.87 a	8.58 ± 0.48 bc	9.34 ± 0.69 ab	7.47 ± 0.67 c
14	His	86.57 ± 2.85 a	77.37 ± 1.96 b	52.57 ± 2.10 c	45.39 ± 1.26 d
15	Lys	11.84 ± 1.57 a	6.16 ± 0.45 b	7.12 ± 0.77 b	5.24 ± 0.74 b
16	Arg	5.07 ± 0.70 a	4.94 ± 0.71 a	3.68 ± 0.34 b	5.36 ± 0.42 a
17	Pro	-	-	5.03 ± 0.23 a	4.89 ± 0.19 b

Note: Values with different letters (a–d) within the same line were significantly different (*p* < 0.05). “-”: not detected.

**Table 3 molecules-27-08857-t003:** The average scores of each sensory attribute in different oils.

Samples	Toasted Sesame Aroma	Caramel Aroma	Aroma Persistence	Off-Flavor	Overall Satisfaction
C-SO	5.54 ± 0.78 c	2.76 ± 0.12 d	4.65 ± 0.65 e	3.66 ± 0.24 a	5.65 ± 0.84 d
PH-SO	5.78 ± 0.65 c	2.83 ± 0.35 d	4.88 ± 0.54 d	3.74 ± 0.36 a	5.82 ± 0.57 cd
GPH-SO	5.65 ± 0.53 c	3.45 ± 0.48 c	5.21 ± 0.38 c	2.32 ± 0.25 c	5.93 ± 0.68 c
PHM-SO	6.42 ± 0.69 b	4.51 ± 0.46 b	6.23 ± 0.86 b	2.65 ± 0.46 b	6.21 ± 0.75 b
GPHM-SO	7.12 ± 1.02 a	5.73 ± 0.63 a	6.56 ± 0.75 a	1.82 ± 0.22 d	7.08 ± 0.85 a

Note: Sensory attributes of samples are listed as an average score (*n* = 3, 3 replications with 10 panelists). Duncan’s method of multiple comparison test was used to indicate significant differences (*p* < 0.05) between various sensorial attributes (a–e).

**Table 4 molecules-27-08857-t004:** Comparison of volatile organic compound content of sesame oil with different treatments and sold sesame oil.

No.	Chemical Compound	^1^ Kis	Content (μm/mL)						^2^ Odors
			C-SO	PH-SO	GPH-SO	PHM-SO	GPHM-SO	S-SO	
	**Pyrazines**								
1	Methylpyrazine	781	13.48 ± 0.12 c	24.41 ± 0.13 c	26.02 ± 0.21 c	29.98 ± 0.47 b	32.38 ± 0.29 a	28.28 ± 0.24 b	chocolate, meaty
2	2,5-Dimethylpyrazine	894	3.44 ± 0.08 d	13.63 ± 0.23 c	13.52 ± 0.29 c	16.26 ± 0.65 b	18.23 ± 0.37 a	16.56 ± 0.32 b	potato-like
3	2,6-Dimethylpyrazine	894	10.79 ± 0.67 e	11.19 ± 0.89 d	11.88 ± 0.83 c	13.41 ± 0.73 b	15.11 ± 0.85 a	11.18 ± 0.56 d	roasted, nutty
4	Ethylpyrazine	881	7.27 ± 0.08 a	3.77 ± 0.31 d	4.30 ± 0.16 cd	4.80 ± 0.46 bc	5.29 ± 0.08 b	3.23 ± 0.25 d	peanut buttery
5	2,3-Dimethylpyrazine	894	1.50 ± 0.11 c	2.89 ± 0.26 b	2.88 ± 0.19 b	3.64 ± 0.15 ab	3.86 ± 0.28 a	-	cocoa-like
6	2-Ethyl-5-methylpyrazine	994	9.03 ± 0.24 d	8.80 ± 0.23 d	8.75 ± 0.53 d	10.56 ± 0.28 b	11.74 ± 0.66 a	9.80 ± 0.33 c	grassy
7	Trimethyl-pyrazine	1008	8.06 ± 0.32 e	10.39 ± 0.37 d	10.49 ± 0.18 d	12.48 ± 0.58 b	13.03 ± 0.18 a	11.29 ± 0.47 c	roasted
8	3-Ethyl-2,5-dimethylpyrazine	1107	4.85 ± 0.17 d	6.33 ± 0.53 c	6.15 ± 0.09 c	7.55 ± 0.56 b	8.41 ± 0.32 a	7.38 ± 0.64 b	roasted
9	2,3-Dimethyl-5-ethylpyrazine	1102	-	-	-	-	-	5.23 ± 0.52 a	roasted, nutty
10	2-Ethenyl-6-methylpyrazine	984	0.52 ± 0.02 d	0.73 ± 0.02 c	0.72 ± 0.03 c	1.03 ± 0.04 b	1.15 ± 0.06 a	-	-
11	2-Acetyl-3-methylpyrazine	1130	5.78 ± 0.14 a	3.49 ± 0.27 d	3.34 ± 0.47 d	4.04 ± 0.28 c	4.56 ± 0.25 b	4.23 ± 0.41 bc	grain-roasted
	**Furans**								
12	2-Pentyl-furan	1040	0.15 ± 0.02 c	0.11 ± 0.01 d	0.30 ± 0.02 a	0.31 ± 0.03 a	0.27 ± 0.01 b	0.25 ± 0.01 b	fruity
13	1-(2-Furanyl)-ethanone	878	1.75 ± 0.08 b	1.49 ± 0.29 cd	1.55 ± 0.08 c	1.78 ± 0.09 b	1.92 ± 0.09 a	1.45 ± 0.16 d	coffee aroma
14	2-Furanmethanol, acetate	1009	0.46 ± 0.01 d	0.56 ± 0.04 c	0.58 ± 0.04 c	0.63 ± 0.06 b	0.72 ± 0.05 a	-	-
15	2-Furanmethanol	885	1.32 ± 0.06 d	2.54 ± 0.08 c	2.51 ± 0.05 c	2.99 ± 0.12 b	3.42 ± 0.14 a	2.58 ± 0.21 c	cooked sugar
16	2-Acetylfuran	925	-	-	-	-	-	1.65 ± 0.12 a	sweet, popcorn
17	5-Methyl-2-furancarboxaldehyde	920	5.28 ± 0.14 c	5.44 ± 0.28 c	5.43 ± 0.29 c	6.42 ± 0.42 b	7.09 ± 0.18 a	5.68 ± 0.24 c	caramel-like
18	2-Acetyl-5-methylfuran	967	0.33 ± 0.02 c	0.34 ± 0.02 c	0.34 ± 0.01 c	0.42 ± 0.02 b	0.49 ± 0.01 a	-	nutty aroma
	**Sulfur compounds**								
19	4-Methylthiazole	832	-	-	-	0.84 ± 0.02 b	0.94 ± 0.03 a	-	nutty, green
20	2,4-Dimethylthiazole	922	-	0.49 ± 0.02 d	0.48 ± 0.02 d	0.60 ± 0.03 c	0.72 ± 0.01 b	0.82 ± 0.06 a	like garlic
21	Dimethyl trisulfide	972	0.67 ± 0.02 d	0.87 ± 0.02 c	0.96 ± 0.02 b	1.10 ± 0.06 a	1.14 ± 0.05 a	0.78 ± 0.05 c	fresh onion
22	2-Methyl-2-thiazoline		-	-	-	-	-	1.16 ± 0.08 a	sulfurous
23	1-(2-Thienyl)-ethanone	1030	0.57 ± 0.02 a	0.37 ± 0.02 e	0.40 ± 0.01 d	0.48 ± 0.02 c	0.53 ± 0.01 b	-	Sulfurous, nutty
	**Pyrroles**								
24	1-Methyl-1H-pyrrole-2-carboxaldehyde	1054	1.73 ± 0.15 d	1.74 ± 0.13 d	1.67 ± 0.12 e	1.89 ± 0.10 c	2.25 ± 0.14 a	2.12 ± 0.16 b	-
25	1-Ethyl-1H-pyrrole-2-carboxaldehyde	955	0.73 ± 0.07 b	0.61 ± 0.06 c	0.59 ± 0.04 c	0.73 ± 0.04 ab	0.86 ± 0.06 a	0.79 ± 0.05 a	-
26	1-(1H-Pyrrol-2-yl)-ethanone	1035	2.81 ± 0.17 a	1.88 ± 0.13 d	1.66 ± 0.13 e	2.17 ± 0.19 c	2.52 ± 0.21 b	2.26 ± 0.14 c	walnut
27	1H-Pyrrole-2-carboxaldehyde	988	5.08 ± 0.08 c	5.00 ± 0.04 c	4.95 ± 0.15 c	6.05 ± 0.19 b	6.62 ± 0.21 a	5.25 ± 0.15 c	-
28	Methyl pyrrole-2-carboxylate	1066	0.33 ± 0.04 b	0.14 ± 0.01 b	0.12 ± 0.01 b	0.15 ± 0.02 b	0.16 ± 0.02 a	-	-
	**Pyridines**								
29	2-Methyl-pyridine	787	-	0.35 ± 0.03 c	0.54 ± 0.02 b	0.27 ± 0.01 c	0.71 ± 0.08 a	0.57 ± 0.01 b	unpleasant
30	1-(2-Pyridinyl)-ethanone	1023	0.50 ± 0.05 a	0.38 ± 0.03 c	0.36 ± 0.02 c	0.49 ± 0.04 ab	0.46 ± 0.03 b	0.59 ± 0.04 a	tobacco-like
31	Methyl nicotinate	1054	0.82 ± 0.06 a	0.47 ± 0.03 d	0.46 ± 0.04 d	0.58 ± 0.05 c	0.69 ± 0.06 b	0.27 ± 0.05 e	caramellic nutty
	**Aldehydes**								
32	Hexanal	806	-	-	-	-	0.88 ± 0.05 a	-	fruity, woody
33	Furfural	831	6.81 ± 0.25 a	5.22 ± 0.21 c	5.44 ± 0.24 c	6.10 ± 0.12 b	6.79 ± 0.13 a	5.14 ± 0.31 c	almond-like
34	Benzaldehyde	982	1.71 ± 0.11 d	1.79 ± 0.10 d	1.81 ± 0.08 d	2.18 ± 0.16 b	2.40 ± 0.27 a	1.92 ± 0.12 c	almond-like
35	(E)-2-Octenal	1013	1.14 ± 0.04 d	1.24 ± 0.02 bc	1.19 ± 0.05 cd	1.33 ± 0.07 ab	1.37 ± 0.05 a	1.09 ± 0.08 d	cooked rice
36	Heptanal	917	-	-	-	-	-	0.78 ± 0.06 a	fatty, rancid
	**Ketones**								
37	3-Octen-2-one	960	-	0.03 ± 0.01 b	-	0.03 ± 0.01 b	0.12 ± 0.02 a	-	fruity, lemon
38	Isophorone	1097	-	0.12 ± 0.01 b	0.12 ± 0.02 b	0.14 ± 0.01 a	0.16 ± 0.01 a	0.15 ± 0.01 a	like camphor
39	1,4-Cyclohex-2-enedione	1044	0.26 ± 0.02 a	0.20 ± 0.01 b	0.19 ± 0.02 b	0.24 ± 0.02 b	0.29 ± 0.02 a	-	-
40	Acetophenone	1029	1.13 ± 0.03 d	1.60 ± 0.08 c	1.57 ± 0.05 c	1.84 ± 0.06 b	2.12 ± 0.11 a	1.75 ± 0.11 b	oranges
41	2-Octanone	1007	-	-	-	-	-	0.56 ± 0.04 a	soapy, buttery
	**Phenols**								
42	2- Methoxy-phenol	1090	8.79 ± 0.14 d	9.62 ± 0.19 c	8.93 ± 0.11 d	11.18 ± 0.15 b	12.81 ± 0.28 a	7.56 ± 0.25 e	smoky
43	Phenol	901	0.61 ± 0.03 a	0.45 ± 0.03 c	0.42 ± 0.02 c	0.54 ± 0.04 b	0.63 ± 0.05 a	0.53 ± 0.04 b	fried meat
44	4-Ethyl-2-methoxyphenol	1303	0.51 ± 0.04 a	0.30 ± 0.02 c	0.27 ± 0.03 c	0.36 ± 0.02 b	0.29 ± 0.03 c	-	slightly sweet
45	2-Methoxy-4-vinylphenol	1293	0.96 ± 0.07 b	0.81 ± 0.04 c	0.68 ± 0.05 d	0.91 ± 0.06 b	1.17 ± 0.08 a	0.88 ± 0.06 c	roasted peanut
46	2,3-Methylenedioxyphenol	1245	0.98 ± 0.09 c	1.42 ± 0.08 b	1.04 ± 0.06 c	1.44 ± 0.07 b	1.89 ± 0.06 a	1.53 ± 0.05 b	-
	**Alcohols**								
47	1-Octen-3-ol	969	1.12 ± 0.06 c	1.00 ± 0.08 c	1.06 ± 0.07 c	1.22 ± 0.05 b	1.37 ± 0.05 a	1.23 ± 0.05 b	strong earthy
48	1-Octanol	1059	-	0.28 ± 0.02 b	0.33 ± 0.07 ab	0.34 ± 0.03 ab	0.38 ± 0.03 a	0.24 ± 0.03 b	orange–rose
49	Benzyl alcohol	1036	0.18 ± 0.02 b	0.17 ± 0.02 b	0.15 ± 0.01 b	0.18 ± 0.02 b	0.23 ± 0.01 a	0.14 ± 0.02 b	slightly sweet

Note: Data are presented as mean ± SD and represent mean of three independent replicates. Values with different letters (a–e) within the same row of oils were significantly different (*p* < 0.05). “-” means not detected. ^1^ Kovats indices (KI) were determined by searching the mass spectrum in the database and manual interpretation. ^2^ Odors indicate the odor bias of some specific flavor compounds.

**Table 5 molecules-27-08857-t005:** Odor activity values (OAVs) of main volatile compounds in different oils.

No.	Chemical	Odor Threshold (mg/m^3^)	OAVs					
	Compound		C-SO	PH-SO	GPH-SO	PHM-SO	GPHM-SO	S-SO
1	Methyl-pyrazine	27	0.50	0.90	0.96	1.11	1.20	1.05
2	2,5-Dimethyl-pyrazine	2.6	1.32	5.24	5.20	6.25	7.01	6.37
3	2,6-Dimethyl-Pyrazine	1.021	10.57	10.96	11.64	13.13	14.80	10.95
4	2,3-Dimethyl-pyrazine	0.123	12.20	23.50	23.41	29.59	31.38	-
5	2-Ethyl-5-methyl-pyrazine	0.32	28.24	27.53	27.36	33.03	36.71	30.63
6	Trimethyl-pyrazine	0.29	27.82	35.84	36.18	43.05	44.94	38.93
7	3-Ethyl-2,5-dimethyl-pyrazine	0.024	202.10	263.80	256.30	314.60	350.40	307.50
8	2-Pentyl-furan	0.1	1.50	1.20	3.10	3.20	2.80	2.50
9	4-Methylthiazole	0.055	-	-	-	15.27	17.09	-
10	Dimethyl-trisulfide	0.0025	268.00	348.00	384.00	440.00	460.00	328.00
11	Hexanal	0.08	-	-	-	-	11.00	-
12	Benzaldehyde	0.06	28.57	29.67	30.17	36.17	39.83	32.06
13	2- Methoxy-phenol	3	2.93	3.21	2.98	3.72	4.27	2.52
14	1-Octen-3-ol	0.001	1120.23	1000.28	1061.13	1217.51	1367.56	1230.80

Note: The odor threshold of each compound was divided by odor thresholds in oil published in “Compilations of odour threshold values in air, water and other media”. “-” means not detected.

**Table 6 molecules-27-08857-t006:** Fitting equation between peroxide value (y) and storage time (x) during 60 ± 3 °C accelerated oxidation.

Oil Samples	Logistic Fitting Curve	R2	Forecast Storage Time (d)
C-SO	$y=9.2410\;-\;9.1542/(1+{\left(\frac{x}{24.1872}\right)}^{2.4019})$	0.996	31
PH-SO	$y=8.5422\;-\;8.4395/(1+{\left(\frac{x}{23.6567}\right)}^{2.4956})$	0.997	33
GPH-SO	$y=8.9512\;-\;8.5894/(1+{\left(\frac{x}{25.2959}\right)}^{2.4781})$	0.998	33
PHM-SO	$y=6.5713\;-\;6.4696/(1+{\left(\frac{x}{22.3525}\right)}^{2.5347})$	0.995	53
GPHM-SO	$y=7.7383\;-\;7.6477/(1+{\left(\frac{x}{32.5284}\right)}^{1.9067})$	0.988	60

Note: The fitting equation was fitted by nonlinear logistic curve function. The forecast storage time is the earliest possible time when the peroxide value of sesame oil does not meet the standard, where the standard refers to the Chinese national standard GB/T 8233-2018.

## Data Availability

The data that support the findings of this study are available from the corresponding author upon reasonable request.

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
