# Peer review of "Application of Maillard Reaction Products Derived Only from Enzymatically Hydrolyzed Sesame Meal to Enhance the Flavor and Oxidative Stability of Sesame Oil"

_molecules, 2022, doi:10.3390/molecules27248857_

Round 1

Reviewer 1 Report

Dear authors,

This paper deals with the Application of Maillard reaction products derived from enzymatically hydrolyzed sesame meal to enhance the flavor and oxidative stability of sesame oil. There are some issues that must be addressed before being considered for publication.

Abstract. Consider adding more quantitative data.

Introduction section, the authors provide enough information to support the study.

Material section Consider providing all chemicals companies' names, equipment brand names, companies, countries, and cities.

Line 119.. All the Maillar products stay in the liquid phase or the solid phase (if any)? please, clarified this point in the manuscript.

Line 138-156... Provide separate methodologies for each method, it is not clear in the way it was presented.

Line 157.. which compound has the formula..C2HCl3O2

results and discussion section.

Line 282--Table1

Figures....check if the Figures have abbreviations or not. Fig or Figure?. Also in Fig. 2. The legends from graphs A and B must be placed in a different position, but not aside, It can be better if placed in vertical order because at first glance they can be confusing.

Line 545..figure 4 is missing in the text.

In general, need more discussion and comparison with other authors' published results for all parameters.

You performed a lot of statistic analysis but no mention in the body text was made..

Too large footnotes for each Table are presented, please consider to shorten.

Author Response

Dear Reviewer #1,

Many thanks to your advice and recognition of the manuscript. The authors have read each suggestion carefully and have modified and improved our manuscript according to your kind advices and detail suggestions.

Abstract.

Comment 1: “Consider adding more quantitative data.”

Response: We sincerely appreciate the valuable comments. In our resubmitted manuscripts, we have modified this section. The specific modifications and locations were in lines 21-25 “The composition of pyrazines (119.35 μg/mL), furans (13.95 μg/mL), and sulfur substances (6.25 μg/mL) contributed positively to sensory properties in GPHM-SO, and 2,5-dimethylpyrazine, 2,6-dimethylpyrazine, and 2,3-dimethylpyrazine were characterized as the key flavor compounds with odor activity values of 7.01, 14.80, 31.38, respectively.

Introduction section

Comment 2: “the authors provide enough information to support the study.”

Response: Thanks for your great suggestion on improving the accessibility of our manuscript. We have added some information from the references to support our study. The specific modifications and locations were as follows.

(1) Lines 46-52: “Specifically, methanethiol, dimethyl disulfide, methyl-pyrazine, 2-ethyl-6-methyl-pyrazine, 3-ethyl-2, 5-dimethyl-pyrazine, 2-furan methanol, etc. would play important roles in the flavor of sesame oil [7]. Meanwhile, as reported, pyrazine and other substances might be the main ingredients of sesame oil essence [8]. Recently, the cheap high temperature treated sesame oils was found to be illegally added to the essence to create high-flavor sesame oil, and became to be a challenge for the public health [9].”.

(2) Lines 67-71: “Shen et al. [15] used different sulfur-containing substances and defatted sesame meal to produce Maillard reaction products with high flavor acceptance, which could be used to improve the flavor of specific foods. However, there is no information about only using sesame meal to produce MR and further improve the flavor of sesame oil.”.

Material section

Comment 3: “Consider providing all chemicals companies' names, equipment brand names, companies, countries, and cities.”

Response: We sincerely appreciate the valuable comments. We have checked the manuscript material section and added the chemicals companies' names, equipment brand names, companies, countries, and cities of origin after the reagent with the instrument. The specific modifications and locations were as follows.

(1) Lines 98-99: “All other solvents and reagents with analytical-grade purity were provided by Jiangsu Jincheng Reagent Co., Ltd. (Jiangsu, China)”.

(2) Lines 123-124: “Then, the hydrolysate was centrifuged at 8000 rpm for 20 min using a high-speed centrifuge (TGL16M, Yancheng KaiT Experimental Instrument Co., Ltd., Jiangsu, China)”.

(3) Lines 130-132: “The hydrolysates were transferred into an oil bath and heated in a thermostatic oil bath (DF-101Z, Shanghai LICHEN-BX Instrument Technology Co., Ltd., Shanghai, China) at 120 °C for 1 h with magnetic stirring at 800 r/min.”.

(4) Lines 152-154: “After the mixture was cooled to room temperature, ultraviolet–visible spectrophotometer (UV-4802, Unico Shanghai Instrument Co., Ltd. Shanghai, China) was used to determine the absorbance of the mixture at 540 nm”.

(5) Lines 211-212: “Then, the volatile substances were adsorbed by the DVB-CAR-PDMS extraction head (Supelco Co., Bellefonte, USA), which was aged for 1 h at 250 °C for the first time”.

(6) Lines 245-247: “The vials were placed in an electrothermal incubator (DHP-48/95B, Tianjin Sedelis experimental and Analytical Instrument Manufacturing Co., Ltd., Shenzhen, China) at 60±3 ℃ for 25 days to accelerate their oxidation”.

Comment 4: “Line 119. All the Maillar products stay in the liquid phase or the solid phase (if any)? please, clarified this point in the manuscript.”

Response: We sincerely appreciate the valuable comments. The Maillard products were mainly stay in the liquid phase. Based on your suggestions, we have modified this section. The specific modifications and locations were in lines 132-134 “At the end of the reaction, the Maillard products liquid of PH and GPH were denoted as PHM and GPHM, respectively.”.

Comment 5: “Lines 138-156... Provide separate methodologies for each method, it is not clear in the way it was presented.”

Response: Thanks for your great suggestion on improving the accessibility of our manuscript. We have added the separate methodologies for sugar content determination in lines 150-162, as following:

Secondly, the pretreated sample (2 mL) was mixed with DNS (2 mL) reagent (with de-ionized water as the control) and reacted at 90 °C for 6 min. After the mixture was cooled to room temperature, ultraviolet–visible spectrophotometer (UV-4802, Unico Shanghai In-strument Co., Ltd. Shanghai, China) was used to determine the absorbance of the mixture at 540 nm. Finally, the contents of reducing sugar and total sugar were determined ac-cording to the standard glucose curve (y=8.9174X-0.0118, R2=0.991, Figure S1).

For determination of total sugar, the sample (5 mL) was mixed with HCl (10 mL, 6mol/L) in a boiling water bath for 10 min, and the pH was adjusted to 7.0 with NaOH (6 mol/L) after the mixture was cooled to room temperature, then the volume was fixed to 25 mL with deionized water in the total sugar assay. The pretreated sample (2 mL) was mixed with DNS, the subsequent procedure was consistent with that of reducing sugar determination.

Comment 6: “Line 157. which compound has the formula. C2HCl3O2.”

Response: We sincerely appreciate the valuable comments. In our resubmitted manuscript, this section has been revised. The specific modifications and locations were in line 170-171 “The sample (after freeze-drying 0.04 g) was dissolved in 10% trichloroacetic acid (6 mL) and placed at 4 °C overnight.

Results and discussion section.

Comment 7: “Line 282--Table1 Figures....check if the Figures have abbreviations or not. Fig or Figure? Also in Fig. 2. The legends from graphs A and B must be placed in a different position, but not aside. It can be better if placed in vertical order because at first glance they can be confusing.”

Response: We think this is an excellent suggestion. In our resubmitted manuscript, we changed the abbreviation “Fig” to the full name “Figure”. The Figures A and B have been arranged in a vertical order, were in lines 328-329. Thanks for your suggestion.

Comment 8: “Line 545. figure 4 is missing in the text.”

Response: In our resubmitted manuscripts, Figure 4 have been added in line 565-566. Thanks for your correction.

Comment 9: “In general, need more discussion and comparison with other authors' published results for all parameters.”

Response: We think this is an excellent suggestion. In our resubmitted manuscripts, we have added the discussion and comparison with other authors' published results. The specific modifications and locations were as follows.

(1) Lines 299-300: “Zhu et al. [23] found that histidine was one of the main sources of pyrazine in Maillard reaction.”.

(2) Lines 442-445: “The content of nitrogen-containing heterocycles and some special pyrazines increased after the oil treated with GPHM, which might was the improvement of the overall flavor of the resulting oil. The changes of pyrazines were consistent with sensory analysis and previous results [10, 11].”.

(3) Lines 460-463: “Moreover, 5-methyl-2-furfuraldehyde and 2-furan-methanol might be the common furan volatile components in sesame oil and usually have a sweet fragrance. These furans might be the key to increase the caramel aroma of the oil, which was consistent with the results of Yin et al. [18].”.

(4) Lines 478-481: “These sulfides had a great influence on the improvement of the oil flavor. The elevating content of the sulfides might be one of the reasons for the increased aroma abundance of the oils, which was consistent with the reported by Park et al. [26].”.

(5) Lines 515-517: “These substances could be assumed as the key aromatic compounds that were made up of the oil aroma, which was basically consistent with the results reported by Jia et al. [6]”.

(6) Lines 550-552: “The acid value of all the oils changed conspicuously during the storage, indicated that the hydrolysis reaction of the all oils hardly occurred during the accelerated oxidation period, which was consistent with the previous reported results [36].”.

Comment 10: “You performed a lot of statistic analysis but no mention in the body text was made.”

Response: We sincerely appreciate the valuable comments. We have added the application of statistical analysis to the resubmitted manuscript. The specific modifications and locations were as follows.

(1) Lines 293-294: “Obviously, the free amino acids significantly (p < 0.05) decreased after the meal treated by both glucoamylase and MR.”.

(2) Lines 300-302: “The histidine content (45.39 μg/g) in GPHM was significantly (p < 0.05) lower than that of GPH (77.37 μg/g), implying that more histidine in the hydrolysate treated by glucoamylase was involved in the MR.”.

(3) Lines 309-312: “After adding the MRPs of the meal hydrolysate with (without) glucoamylase, a* and b* values of the oils changed non-significantly (p > 0.05), such as a* value varied from 0.31 (C-SO) to 0.36 (GPHM-SO), and b* value reduced from -1.07 (C-SO) to -1.61 (PHM-SO)”.

(4) Lines 312-314: “Moreover, the oil after being treated by the MRPs of the meal hydrolysate with glucoamylase, a* value increased significantly (p < 0.05), which represented an increase of oil redness”.

(5) Lines 319-321: “Compared with other oils, L* values of the both oils significantly (p < 0.05) decreased, indicating that the color of the oils treated with MRPs was darker than that of the other oils”.

(6) Lines 343-345: “Obviously, in all the oils, after being treated by GPHM, the overall satisfaction of the oil (GPHM-SO) was increased significantly (p < 0.05).”.

(7) Lines 427-429: “In addition, MRPs-treated oils (PHM-SO and GPHM-SO) had higher levels of pyrazine than other oils, especially the pyrazine content in GPHM-SO was significantly (p < 0.05) increased”.

(8) Lines 455-456: “Obviously, the total furans were found in the oils treated by GPHM increased significantly (p < 0.05), as there were more monosaccharides and oligosaccharide in sesame meal after being treated by glucoamylase”.

(9) Lines 469-471: “After the addition of the meal hydrolysate and their MRPs in the crude oil, the composition of sulfur compounds increased significantly (p < 0.05), …”.

(10) Lines 486-488: “As shown in Figure S2d, the total content of aldehydes in PH-SO and GPH-SO were 8.25 and 8.44 μg/mL, respectively, and were significantly (p < 0.05) lower than C-SO (11.07 μg/mL)”.

(11) Lines 542-543: “Moreover, after being treated by GPHM, the stability of oil was improved significantly (p < 0.05)”.

Comment 11: “Too large footnotes for each Table are presented, please consider to shorten.”

Response: We sincerely appreciate the valuable comments. We have shortened down the table footnotes, and the exact were as follows.

In Table 3, lines 351-352, we deleted the note 2 “C-SO stands for the crude oil, PH-SO stands for the oil treated by hydrolysate without glucoamylase, GPH-SO stands for the oil treated by hydrolysate with glucoamylase, PHM-SO stands for the oil treated by MRPs of the hydrolysate without glucoamylase and GPHM-SO stands for the oil treated by MRPs of the hydrolysate with glucoamylase.”

In Table 6, lines 573-575, we replace the note with the following: “Note: The fitting equation was fitted by non-linear curve function Logistic. The forecast storage time is the earliest possible time when the peroxide value of sesame oil does not meet the standard where the standard refers to the Chinese national standard GB/T 8233-2018”.

We tried our best to improve the manuscript and made changes marked in red in revised paper. We appreciate for Editors/Reviewers’ warm work earnestly, and hope the correction will meet with approval. Once again, thank you very much for your comments and suggestions.

Reviewer 2 Report

The authors attempted to improve the aroma quality of sesame oil by MRPs (Maillard Reaction Products) that are derived only from sesame meal without exogenous addition. Furthermore, they improved this method to solve the problem of insufficient aroma of the sesame oil caused by a lower temperature (160 ℃) roasting. Specifically, they treated sesame meal hydrolysates with protease and glucoamylase to obtain the amino acids and reducing sugars. Then they processed the MRPs without exogenous additions and investigated the effects of PH, GPH, PHM, and GPHM on the aroma of low-temperature roasting sesame oil. Finally, they found that the oxidative stability of oil was significantly improved with the addition of MRPs. The manuscript is well-written and the presented method is technically sound. However, I do have some suggestions that hopefully can help them to improve the quality of the paper. I am quite open to looking at a revised version if the authors could address some major and minor issues in a satisfactory fashion, which we describe in more detail below.

Major issues:

1.     In the abstract, the authors didn’t capture the research question they intended to solve before moving on to their results. I strongly suggest they briefly define the research questions and current obstacles in the beginning, similar to what they summarized in the first two paragraphs of the introduction section.

2.     I think some notes/captions for tables are positioned in the wrong place. For example, in lines 263-264, they wrote the note “PH stands for the meal treated by without glucoamylase, PHM stands for the MRPs of the hydrolysate without glucoamylase, GPH stands for the meal treated by with glucoamylase and GPHM stands for the MRPs of the hydrolysate with glucoamylase.” I think the note is for table 2 instead of table 1. Please double check all notes carefully.

3.     Figure 3: Please use the same colors and markers for five categories (i.e., S-CO, PH-SO, GPH-SO, PGM-SO, GPHM-SO) for panels A and B.

4.     All citation labels are duplicated. Please de-duplicate the citation labels.

Minor comments:

1.     Line 72: “MRPs” are mentioned without specifying the acronym. I assume it is “Maillard Reaction Products”. Please refer to the full name before using the acronym.

2.     Line 250-251: “The higher sugars would be favor of the production of Maillard products” -> “The higher sugars would be in favor of the production of Maillard products”

3.     Line 295: “Color is an also important indicator for the quality oils.” -> “Color is another important indicator for assessing the quality of oils.”

4.     Line 315: “As shown in Fig. 2B, L* (histogram) values of the PHM-SO was…” -> ““As shown in Fig. 2B, L* (histogram) values of the PHM-SO were…”

5.     Line 328: “Compared to the crude oil, after treated by PH…” -> “Compared to the crude oil, after being treated by PH…”

6.     Line 453: “sesame meal after treated by glucoamylase.” -> “sesame meal after being treated by glucoamylase.”

7.     Line 533: “…the initial oxidation stage of oils as higher value…” -> “…the initial oxidation stage of oils as a higher value…”

8.     Lines 552-553: “As shown in Table 6, the R2 values of all oil sample fitting equations were above 0.95, and the fitting results well.” -> “As shown in Table 6, the R2 values of all oil sample fitting equations were above 0.95, implying great fitting results.”

Author Response

Dear Reviewer #2,

Many thanks to your advice and recognition of the manuscript. The authors have modified and improved our manuscript according to your kind advices and detail suggestions.

Comment: “The authors attempted to improve the aroma quality of sesame oil by MRPs (Maillard Reaction Products) that are derived only from sesame meal without exogenous addition. Furthermore, they improved this method to solve the problem of insufficient aroma of the sesame oil caused by a lower temperature (160 ℃) roasting. Specifically, they treated sesame meal hydrolysates with protease and glucoamylase to obtain the amino acids and reducing sugars. Then they processed the MRPs without exogenous additions and investigated the effects of PH, GPH, PHM, and GPHM on the aroma of low-temperature roasting sesame oil. Finally, they found that the oxidative stability of oil was significantly improved with the addition of MRPs. The manuscript is well-written and the presented method is technically sound. However, I do have some suggestions that hopefully can help them to improve the quality of the paper. I am quite open to looking at a revised version if the authors could address some major and minor issues in a satisfactory fashion, which we describe in more detail below.”

Response: Thank you very much for your time involved in reviewing the manuscript and your very encouraging comments on the merits. We also appreciate your clear and detailed feedback and hope that the explanation has fully addressed all of your concerns. In the remainder of this letter, we discuss each of your comments individually along with our corresponding responses.

Major issues:

Comment 1. “In the abstract, the authors didn’t capture the research question they intended to solve before moving on to their results. I strongly suggest they briefly define the research questions and current obstacles in the beginning, similar to what they summarized in the first two paragraphs of the introduction section.”

Response: Thanks for your great suggestion on improving the accessibility of our manuscript. We have added the research questions and current obstacles in the beginning. The specific modifications and locations were in lines 12-13 “The low temperature roasting sesame oil have become increasingly popular because of the nutrition, however, the flavor would be reduced”.

Comment 2. “I think some notes/captions for tables are positioned in the wrong place. For example, in lines 263-264, they wrote the note “PH stands for the meal treated by without glucoamylase, PHM stands for the MRPs of the hydrolysate without glucoamylase, GPH stands for the meal treated by with glucoamylase and GPHM stands for the MRPs of the hydrolysate with glucoamylase.” I think the note is for table 2 instead of table 1. Please double check all notes carefully.”

Response: The reviewer is right, Actually, the contents in the tables should be revised according to the notes, and the revisions have been marked in the red color now, by the way, all the notes of the tables have been checked now.

Comment 3. “Figure 3: Please use the same colors and markers for five categories (i.e., S-CO, PH-SO, GPH-SO, PGM-SO, GPHM-SO) for panels A and B.”

Response: We think this is an excellent suggestion. In our resubmitted manuscripts, the colors in Figure 3 panels A and B have been harmonized.

Thanks for your suggestion again.

Comment 4. “All citation labels are duplicated. Please de-duplicate the citation labels.”

Response: In our resubmitted manuscripts, duplicate citation labels have been removed. Thanks for your correction.

Minor comments:

Comment 5. “Line 72: “MRPs” are mentioned without specifying the acronym. I assume it is “Maillard Reaction Products”. Please refer to the full name before using the acronym.”

Response: In our resubmitted manuscript, the full name “Maillard Reaction Products” for “MRPs” has been supplemented where it first. Thanks for your correction.

Comment 6. “Line 250-251: “The higher sugars would be favor of the production of Maillard products” -> “The higher sugars would be in favor of the production of Maillard products”.”

Response: We sincerely appreciate the valuable comments. We have revised this part based on your comments.

Comment 7. “Line 295: “Color is an also important indicator for the quality oils.” -> “Color is another important indicator for assessing the quality of oils.”.”

Response: We think this is an excellent suggestion. We have revised this part based on your comments as “Color is another important indicator for assessing the quality of oils.”

Comment 8. “Line 315: “As shown in Fig. 2B, L* (histogram) values of the PHM-SO was…” -> ““As shown in Fig. 2B, L* (histogram) values of the PHM-SO were…”.”

Response: Thanks for your careful checks. Based on your suggestions, we have modified this section as “As shown in Fig. 2B, L* (histogram) values of the PHM-SO were 54.60 and GPHM-SO were 54.58.

Comment 9. “Line 328: “Compared to the crude oil, after treated by PH…” -> “Compared to the crude oil, after being treated by PH…”.”

Response: Thank you for your suggestion, this section has been revised as “Compared to the crude oil, after being treated by PH…

Comment 10. “Line 453: “sesame meal after treated by glucoamylase.” -> “sesame meal after being treated by glucoamylase.”.”

Response: Thank you for your suggestion. In our resubmitted manuscript, this section has been revised as “sesame meal after being treated by glucoamylase.

Comment 11. “Line 533: “…the initial oxidation stage of oils as higher value…” -> “…the initial oxidation stage of oils as a higher value…”.”

Response: Thanks, we have modified this section as “…the initial oxidation stage of oils as a higher value…

Comment 12. “Lines 552-553: “As shown in Table 6, the R2 values of all oil sample fitting equations were above 0.95, and the fitting results well.” -> “As shown in Table 6, the R2 values of all oil sample fitting equations were above 0.95, implying great fitting results.”.”

Response: Thank you for your suggestion. We have made the modification as “As shown in Table 6, the R2 values of all oil sample fitting equations were above 0.95, implying great fitting results.

We have tried our best to improve the manuscript and made some changes marked in red in revised paper. We appreciate for Editors/Reviewers’ warm work earnestly, and hope the correction will meet with approval. Once again, thank you very much for your comments and suggestions.

Round 2

Reviewer 1 Report

No further comments

Reviewer 2 Report

The authors answer all my concerns very well. I have no other comments.